# Modern Surgical Techniques of Thyroidectomy and Advances in the Prevention and Treatment of Perioperative Complications

**DOI:** 10.3390/cancers15112931

**Published:** 2023-05-26

**Authors:** Bartłomiej Ludwig, Maksymilian Ludwig, Anna Dziekiewicz, Agnieszka Mikuła, Jakub Cisek, Szymon Biernat, Krzysztof Kaliszewski

**Affiliations:** Department of General, Minimally Invasive and Endocrine Surgery, Wroclaw Medical University, Borowska Street 213, 50-556 Wroclaw, Poland; maksymilian.ludwig@student.umw.edu.pl (M.L.); anna.dziekiewicz@student.umw.edu.pl (A.D.); agnieszka.mikula@student.umw.edu.pl (A.M.); jakub.cisek@student.umw.edu.pl (J.C.); szymon.biernat@student.umw.edu.pl (S.B.)

**Keywords:** thyroidectomy, trans-axillary approach, bilateral axillo-breast approach, transoral endoscopic thyroidectomy via vestibule access, thermal ablation, autofluorescence, indocyanine green, parathyroid identification, perioperative bleeding, recurrent laryngeal nerve, artificial intelligence

## Abstract

**Simple Summary:**

The aim of this study was to collate the achievements to date in the field of the treatment and prevention of complications during thyroidectomy, with a particular focus on the latest findings. The points of interest were modern surgical techniques, the prevention of damage to the parathyroid glands and the recurrent laryngeal nerves during surgery and perioperative bleeding.

**Abstract:**

Thyroid cancer is the most common cancer of the endocrine system, and, in recent years, there has been a phenomenon of overdiagnosis followed by subsequent overtreatment. This results in an increasing number of thyroidectomy complications being faced in clinical practice. In this paper, we present the current state of knowledge and the latest findings in the fields of modern surgical techniques, thermal ablation, the identification and assessment of parathyroid function, recurrent laryngeal nerve monitoring and treatment and perioperative bleeding. We reviewed 485 papers, from which we selected 125 papers that are the most relevant. The main merit of this article is its comprehensive view of the subject under discussion—both general, concerning the selection of the appropriate method of surgery, and particular, concerning the selection of the appropriate method of prevention or treatment of selected perioperative complications.

## 1. Introduction

Thyroid cancer is the most common malignancy of the endocrine system, and the number of cases has been steadily increasing in recent years [1,2,3]. This trend is likely to continue in the coming years, as with the development of technology and diagnostic capabilities, we are increasingly faced with the phenomenon of overdiagnosis. This is undoubtedly one of the biggest near-term challenges for global public health [4,5,6]. Along with the increase in overdiagnosis, there is another phenomenon, overtreatment, which has led to an increase in the number of thyroidectomy procedures being performed [7,8]. Papillary thyroid cancer (PTC) is mainly “responsible” for the above phenomenon and is detected more often every year, while the number of diagnosed cases of follicular, medullary and anaplastic cancers remains relatively stable [9].

According to the American Thyroid Association (ATA)’s management guidelines, patients with differentiated thyroid cancer (DTC)—which includes PTC—greater than 4 cm or with one of the following features should undergo a total or near-total thyroidectomy: T4, N1 or M1 (according to the TNM classification). In patients who do not meet the above conditions but have cancer larger than 1 cm, a lobectomy may be considered in addition to the above procedures if the patient’s clinical condition supports it. If the thyroid cancer does not meet the above criteria and is smaller than 1 cm, a lobectomy procedure should be considered first if there is no clinical indication to extend the procedure [10]. The European Society for Medical Oncology (ESMO)’s guidelines recommend a total thyroidectomy for DTC larger than 4 cm or with an N1 trait according to the TNM classification. If DTCs do not have the characteristics mentioned above, then depending on other risk factors, one of the following procedures should be performed: a total thyroidectomy, a lobectomy or active surveillance [11].

Thyroidectomy, like any other procedure, is not free of postoperative and intraoperative complications. Depending on the method by which a thyroidectomy is performed, various complications can be encountered. The most important complications include recurrent laryngeal nerve palsy, hypoparathyroidism and haematoma [12,13]. With alternative surgical techniques (e.g., the transoral endoscopic vestibular approach), fewer standard complications such as CO_2_ embolism, emphysema, mental nerve injury or skin perforation may be encountered [12].

The purpose of our work was to bring together the achievements to date in the thyroidectomy field with a special focus on perioperative complications and modern options for their prevention and treatment. Our goal was to give a complete understanding of the topic. We discuss different approaches for performing the procedure in the first part of the paper and focus on particular issues during thyroidectomy in the second part. We include the latest developments in this field, such as the new approach (the gasless trans-axillary and modified bilateral axillo-breast approach, transoral endoscopic thyroidectomy via vestibule access (TOETVA)), thermal ablation, autofluorescence imaging and indocyanine green fluorescence. These seem to be the most interesting and most future-oriented developments. We also collect and succinctly summarise the already-known and somewhat older methods of preventing and treating the complications of thyroidectomy, such as perioperative bleeding and laryngeal nerve paresis.

## 2. Materials and Methods

We searched the Google Scholar and PubMed online databases using various combinations of phrases and titles containing the following terms: “thyroidectomy”, “open approach”, “TAA”, “BABA”, “TOETVA”, “thyroid cancer”, “thermal ablation”, “perioperative bleeding”, “laryngeal nerve injury”, “intraoperative neuromonitoring”, “artificial intelligence”, “autofluorescence” and “indocyanine green”. Four authors searched the database and selected the articles to be included in the paper, which were then approved by two other authors. In total, we analysed 485 articles, from which we finally selected 125 that we considered the most relevant to our work. The process and criteria for inclusion of papers in our review are shown in the diagram in Figure 1.

## 3. Results

### 3.1. Modern Surgical Techniques

#### 3.1.1. Introduction

The surgical techniques used in a thyroidectomy are a crucial part of the treatment of thyroid diseases. For many years, surgeons have used various methods to safely remove all or part of the thyroid gland [14]. Traditionally, thyroid surgery is performed using the open surgery approach, which involves making a large anterior neck incision and removing the thyroid gland through the opening [15]. The transcervical incision provides excellent visualisation and very good access to the thyroid gland, parathyroids and the recurrent laryngeal nerves (RLNs) as well as great control over the surgical field [16,17]. The conventional open approach is the standard procedure for thyroid cancer, but it can result in a visible scar on the neck. This can greatly affect a patient’s quality of life due to reduced self-confidence [18,19,20].

Considering the cosmetic aspect, the desire to minimise perioperative and postoperative complications and the 2015 ATA guidelines, there is currently a focus on developing alternative, cosmetically favourable surgical methods in thyroid surgery, while eliminating or minimising the anterior neck incision [10]. One focus of these innovations is based on minimally invasive techniques, such as the trans-axillary approach (TAA), the bilateral axillo-breast approach (BABA) and natural orifice surgery. These approaches may offer safer options for selected patients, but they are still relatively new and experimental, so the qualifications and expertise of the surgeon are crucial for ensuring their safety. For a clearer understanding and comparison of these approaches, we provide an overview of the indications, contraindications, benefits, drawbacks and potential advancements of these methods (Table 1). This table aims to provide a comprehensive overview of the various thyroidectomy techniques and the information needed to make informed decisions about the best treatment options for individual patients.

#### 3.1.2. Trans-Axillary Approach (TAA)

Ikeda et al. presented the initial results of trans-axillary thyroid surgery in 2001 [39]. This study demonstrated that the procedure can be successfully performed with a minimal number of complications, thus confirming the feasibility and safety of using the axillary approach. The procedure for performing the technique was clearly outlined in both Ikeda’s original report and other subsequent studies in the literature [38,39,40,41].

Over the past two decades, specific innovations and improvements in TAA have been closely related to the advancements in robotic surgical methods for thyroid tumour treatment. Robotic thyroidectomy (RT), introduced by Miyano in 2008, utilizes surgical robots such as the da Vinci system to facilitate the procedure, offering benefits such as a 3D enlarged view, precise instrument control, accurate joint movement and a gasless technique. RT shares advantages with endoscopic thyroidectomy (ET) [42,43,44,45]. In the domain of thyroid surgery research, extensive investigations have focused on RT due to its superior postoperative outcomes, leading to the development of novel approaches. Among these approaches, the gasless trans-axillary approach (GTAA) has emerged as a unique surgical procedure with the largest experience in oncological thyroid surgery. Notably, GTAA significantly differs from the original CO_2_ insufflation approach. Promising techniques such as the bilateral axillo-breast approach (BABA) and gasless unilateral trans-axillary approach (GUAA) have also contributed to the advancements in thyroid surgery. These ongoing endeavours aim to enhance postoperative outcomes and refine surgical techniques in the field [36,37,46].

GTAA represents a significant advancement in thyroid surgery techniques. This approach, pioneered by Korean surgeons, allows the surgery to be performed without the need for carbon dioxide insufflation. Instead, a special tool is used for lifting and retracting tissue, eliminating the use of gas. The gasless version of TAA has become the preferred approach due to its numerous advantages over the gas-insufflation approach [40,47]. The advantages of the gasless approach include a reduced risk of complications such as subcutaneous emphysema and pneumothorax, better visualisation without gas distorting the view and faster recovery for patients. Unlike gas-insufflated procedures that can cause discomfort and pain after surgery, the gasless technique eliminates the use of gas altogether, resulting in faster patient recovery. A randomised controlled trial by Jantharapattana et al. compared GTAA with open thyroidectomy (OT) in terms of hospitalisation, pain, scar satisfaction, estimated blood loss and complication rates. The study found no significant difference between the two approaches, except for a longer operation duration (297.5 ± 56.2 vs. 156.2 ± 26.9; *p* < 0.001) and higher total treatment costs (USD 940 vs. 454 per case) for GTAA [48]. Compared to OT, GTAA offers better cosmetic outcomes and scar satisfaction measured on a visual analogue scale (VAS) (at six months after surgery: 9.6 ± 0.6 vs. 6.7 ± 1.4; *p* < 0.001), which remains less visible after healing [17,37,48,49]. Many authors have reported satisfactory cosmetic outcomes of remote-access thyroidectomy, but most of them only evaluated short-term subjective cosmetic satisfaction. Lee et al. presented the results of a study comparing postoperative cosmetics in TAA, postauricular facelift (PF) and a conventional transcervical OT. Cosmetic satisfaction was higher in the TAA and PF groups than in the OT group, both three months (VSS scores: 2.36 ± 1.31 vs. 2.42 ± 1.19 vs. 1.52 ± 0.79; *p* < 0.001) and one year (VSS scores: 1.64 ± 0.96 vs. 2.08 ± 1.33 vs. 0.71 ± 0.92) after surgery, with no significant difference in the results between the PF and TAA groups [50]. A meta-analysis presented in 2021 by Jasaitis et al. showed that GTAA is as safe as conventional OT. Taking safety as the primary outcome and comparing conventional and trans-axillary thyroidectomy in terms of overall complication rates, such as postoperative bleeding, vocal cord paralysis, haematoma/seroma formation, hypoparathyroidism, paraesthesia and infection, there was no significant difference in the overall complication rate between the two groups. The advantages of endoscopic GTAA were a lower incidence of hypoparathyroidism, less postoperative pain and a shorter hospital stay. The main disadvantage of this approach was a longer operation duration [17].

Recent research on TAA, published in 2022, focuses on the impact of the procedure on the RLN as well as on the protection of the parathyroids and the management of postoperative pain [51,52,53]. Chen et al. presented the results of a study on the impact of gasless endoscopic thyroidectomy through an axilla (GETTA) on RLN damage in thyroid cancer patients. Clinically, the GETTA group had a shorter operation duration, less intraoperative blood loss, a faster extubation time and a shorter hospitalisation duration compared to the conventional OT group (all *p* < 0.05). The GETTA group also had lower VAS scores at one, two and three days post-surgery compared to the OT group (all *p* < 0.001). In terms of negative emotions, the GETTA group had lower Self-Rating Anxiety Scale and Self-Rating Depression Scale scores compared to the OT group (both *p* < 0.001). The GETTA group also had a lower incidence of postoperative recurrent laryngeal nerve injury compared to the OT group (*p* < 0.001). Additionally, GETTA was found to be a protective factor against RLN injury in thyroid cancer patients (*p* < 0.05). The GETTA group also had a lower Voice Handicap Index and swallowing impairment scores one week and one month post-surgery compared to the OT group (all *p* < 0.001). Lastly, the GETTA group had a lower incidence of postoperative complications such as hypocalcaemia, wound infection and numbness of the hands and feet (all *p* < 0.001) [53].

During thyroid surgery, it is very important to accurately locate the parathyroids to avoid their accidental removal or damage to their blood supply. Several intraoperative procedural techniques, such as indocyanine green (ICG) and nanocarbon (NC) fluorescence, have been studied to distinguish the parathyroids and were found to be safe and feasible [54,55]. It appears that these nanocarbons have a high degree of tropism for the lymphatic system, which means they tend to accumulate in lymphatic tissue. They also have a high tracking speed, fast black staining and high colour contrast with surrounding tissue, which makes them easy to visualise during surgery. Some studies suggest that NC may be used to identify the parathyroids during thyroid surgery, potentially making the procedure more accurate and safer for the patient [56,57]. GETTA with NC and ICG in thyroid cancer may protect the parathyroids and provide satisfactory clinical outcomes.

Chae et al. presented a preliminary retrospective cohort study on the clinical application of intercostal nerve block II to control pain related to thyroid lobe detachment after thyroidectomy through the axilla using a robot [51]. Pectoralis nerve block II (PECS II) could be a useful option for pain management during robotic-assisted trans-axillary thyroidectomy (RATT) and may be used in combination with other methods for pain relief. While further research is needed to determine the optimal duration and technique of PECS II and to identify potential complications or areas of sensory loss, preliminary results suggest that this pain treatment technique may be effective in reducing pain during lobe detachment and improving recovery after surgery. While rare, potential complications of PECS II include infection, injury to the thoraco-brachial artery or pleural space, iatrogenic intravenous injection and local anaesthetic toxicity [58]. Analgesia during RATT may be challenging but is crucial for a successful recovery after surgery. Therefore, further research is being conducted to confirm the pain-relieving effects of PECS II and to develop a pain control scheme for patients undergoing RATT [51].

In conclusion, when considering surgical approaches for treating thyroid tumours, it is important to weigh the advantages and disadvantages of each method. While the trans-axillary approach has been shown to be feasible and safe, it is associated with an increased risk of injury to the RLN and the parathyroid glands. The gasless trans-axillary approach provides many potential benefits, such as better visualisation, faster recovery and better cosmetic outcomes, but it also has a longer operation time.

#### 3.1.3. Bilateral Axillo-Breast Approach (BABA)

In 2007, Choe et al. adapted the axillo-bilateral-breast approach (ABBA) to create the BABA technique for total thyroidectomy. This technique involves the use of an additional contralateral axillary port, which provides optimal visualisation during the procedure [59]. Over the past two decades, the BABA technique has undergone many changes and innovations, including the development of endoscopic and robotic approaches [34,35,60,61,62]. BABA provides the surgeon with a symmetrical view of the thyroid and allows the use of the same methods as are used in OT. Both techniques have been shown to be effective in the removal of the thyroid gland and have similar surgical outcomes [26,61]. However, the benefits and potential drawbacks of each technique may vary depending on the patient and the specific surgical situation. Endoscopic BABA has several limitations, including a two-dimensional view from the camera, its dependence on the operator’s camera skills and non-articulated instrument movement. These limitations make it technically challenging to perform complex surgical procedures using endoscopic instruments, especially for novice surgeons [21,22].

The limitations of endoscopic thyroid surgery have been overcome with the introduction of the da Vinci robotic system. Robot-assisted bilateral axillo-breast approach (BABA–RT) thyroidectomy offers a 15-fold magnification view, a three-dimensional operative field and a stable operative view with the robot arm. In addition, the multi-joint endowrist technology provides better ergonomics, allowing surgeons to perform meticulous surgical procedures more easily. Unlike endoscopic thyroidectomy, the robotic surgical system provides motion filtering and tremor reduction as well as a more versatile articulated instrument. However, BABA–RT using the da Vinci surgical system does require a wider surgical flap and may cause more pain and sensory impairment than other types of minimally invasive techniques [34,35,62]. BABA–RT with a mini-flap was developed to address these drawbacks and is considered a form of minimally invasive thyroidectomy. In 2022, Shin et al. presented a retrospective review of the clinical records of 44 patients who underwent BABA–RT using either a conventional flap or a mini-flap. The new mini-flap procedure, implemented using the da Vinci robot system, shortened the operative time and minimised the surgical flap compared with open thyroidectomy (206.18 ± 31.09 vs. 178.90 ± 34.43 min; *p* = 0.009; 38.85 ± 2.73 vs. 32.21 ± 8.62 min; *p* = 0.003) and improved surgical outcomes by reducing the amount of drainage (196.57 ± 81.40 vs. 150.74 ± 40.80 mL; *p* = 0.027), shortening the hospital stay and reducing the trauma associated with surgical exposure [35].

In comparison to other types of robotic thyroid surgery, BABA provides the largest surgical angles between instruments, leading to sufficient distance between tools to avoid interference [63]. Robotic surgical systems have several advantages over endoscopic procedures in thyroid surgery. They allow for easier manipulation of surgical tools during meticulous procedures, resulting in safe and precise dissection. Robotic systems also help surgeons identify and preserve the parathyroid glands and recurrent laryngeal nerves (RLN).

Lee et al. in 2022 presented a single-surgeon experience of distinctive ET and the lessons learned from each approach. This study considered data on 701 patients who underwent ET via the trans-axillary (TA), BABA, unilateral axillo-breast with carbon dioxide insufflation (UABA), retroauricular (RA) or the transoral approach (TOA) between May 2008 and March 2020. Of the five approaches, the average BABA procedure duration was in the middle, while that of UABA was the shortest (TOA: 196.10 ± 40.19 min; TA: 194.65 ± 51.13 min; BABA: 189.11 ± 61.53 min; RA: 168.22 ± 45.63 min; UABA: 118.62 ± 30.23 min; *p* = 0.02). BABA was the most painful approach, while TOA was the least painful on postoperative day 1 (TA: 3.09 ± 0.96; BABA: 3.59 ± 0.92; UABA: 2.39 ± 0.54; RA: 3.49 ± 0.93; TOA: 2.01 ± 0.37; *p* = 0.04) and day 3 (TA: 2.10 ± 0.77; BABA: 2.59 ± 0.88; UABA: 1.84 ± 0.37; RA: 3.01 ± 0.67; TOA, 1.49 ± 0.45; *p* = 0.04). The best cosmetic results after three months were in the TOA group, followed by BABA (TA: 3.91 ± 1.21; BABA: 4.52 ± 1.13; UABA: 4.49 ± 0.74; RA: 4.28 ± 0.74; TOA: 4.81 ± 0.48; *p* = 0.04) [61].

So far, few reports have compared surgical outcomes between the TOA and BABA approaches. These studies have mainly focused on robotic surgery [64]. In a study presented by Liang et al. in 2021, experiences with endoscopic thyroidectomy using these two approaches were presented [33]. TOA and BABA are used to perform thyroidectomy through small incisions, and they share some common characteristics [24]. Both approaches are designed symmetrically to allow for easy removal of both thyroid lobes, while the TA and RA approaches are unilateral and may be more difficult to apply in cases of thyroid changes on the opposite side of the incision. The TOA and BABA approaches in thyroid surgery use CO_2_ insufflation to create a working space and only require small incisions for trocar insertion. This results in a better cosmetic outcome compared to the TA and RA approaches, which require larger incisions for the insertion of a metal retractor. In terms of surgical outcomes, the TOA group required significantly longer operative durations than the BABA group, both for lobectomy (194.1 ± 35.1 vs. 177.0 ± 40.8 min; *p* = 0.026) and total thyroidectomy (246.0 ± 37.4 vs. 214.3 ± 40.8 min; *p* = 0.042). However, the duration difference became insignificant after the first 20 cases of transoral thyroidectomy. The drainage fluid collected post-surgery was serosanguineous, and the TOA group had a smaller drainage volume than the BABA group (64.9 vs. 78.5 mL; *p* = 0.017). Nevertheless, there was no significant difference in relation to blood loss, hospital stay, postoperative pain evaluation or removed lymph nodes. The frequency of postoperative complications such as parathyroid insufficiency and vocal cord paralysis was comparable in both groups. The results of this study indicated that the surgical outcomes of TOA and BABA were comparable. In contrast, a different study in 2020, which compared postoperative outcomes between BABA–RT and transoral robotic thyroidectomy (TORT), found that the TOA group had a significantly shorter operation duration (204.11 ± 40.19 vs. 243.78 ± 57.16 min; *p* < 0.01). In a study of 248 patients treated with TORT and 316 patients treated with BABA–RT, TORT was found to have not only superior cosmetic outcomes with minimal scarring but also comparable or superior surgical outcomes with a shorter operation duration compared to BABA–RT [32].

In conclusion, BABA has undergone several modifications over the years and now includes endoscopic and robotic approaches. While some studies have found that endoscopic BABA thyroidectomy may result in slightly shorter operating times and less blood loss than robotic BABA thyroidectomy, others have found no significant differences between the two techniques [32,33,34,35,65,66]. The choice of a BABA technique, whether endoscopic or robotic, should be based on the individual patient’s needs and preferences and made on a case-by-case basis by the surgeon. While endoscopic BABA has some limitations, robotic-assisted BABA thyroidectomy offers a three-dimensional operative field, better ergonomics and a 15-fold magnification view. However, BABA–RT requires a wider surgical flap and may cause more pain and sensory impairment than other minimally invasive techniques. BABA provides the largest surgical angles between instruments and has overcome the limitations of endoscopic procedures, allowing for safe and precise dissection while identifying and preserving parathyroids and RLNs. While cosmetic outcomes are generally good, specific benefits and drawbacks may vary depending on the patient and the surgical situation.

#### 3.1.4. Natural Orifice Surgery

Minimally invasive surgery has gained popularity among patients and surgeons, leading to the widespread adoption of the axillary and breast areolar approaches to thyroidectomy. However, some surgeons have questioned the minimally invasive nature of these techniques due to the extensive skin flaps required [67]. In response, Benhidjeb and Wilhelm et al. introduced the first endoscopic transoral thyroidectomy in 2009 [68,69]. This procedure involves intubating the patient through the nose and administering preoperative antibiotics and uses three ports inserted through incisions in the mouth and one sublingual incision for patients with benign thyroid nodules. However, this approach has faced technical difficulties and a high rate of complications [70,71,72]. Anuwong recognised the limitations of this method and proposed the vestibular approach (TOETVA) in 2016, which significantly reduced complications [73]. In 2018, he published the initial outcomes of 400 cases of TOETVA, showing similar surgical outcomes and complication rates to those of OT [74].

Robotic surgery has been used to enhance surgical manoeuvring in the TOA approach to thyroidectomy. TOETVA was found to have significantly lower complication rates and better surgical movement compared to other approaches [30]. While TOA has the potential to be a safer option for selected patients, it is still a relatively new and experimental technique, so the long-term outcomes are not yet well understood [25,75,76,77]. There are also some potential risks and limitations to consider, such as the small working spaces or the possibility of postoperative seroma and infection [31].

The qualifications and expertise of the surgeon, particularly in regard to endoscopic surgery, are crucial for ensuring the safety of the procedure. It is essential that the surgeon is proficient in the standard procedures for central neck surgery, including thyroidectomy, parathyroidectomy and central neck dissection, and is able to perform open surgery if needed. To become proficient in transoral thyroidectomy (TO), it is recommended that surgeons participate in hands-on cadaver training, ideally with guidance from experienced TO surgeons, and perform at least 11 cases. This helps to overcome the learning curve and improve surgical skills [78]. Some researchers believe that more than 50 cases may be necessary to fully overcome the learning curve [31]. In 2020, Lira et al. presented the results of a retrospective cohort study in which the learning curve and early surgical outcomes of TOETVA were assessed and compared to those of conventional thyroidectomy. This study found that TOETVA is a procedure that is safe and feasible, and the learning curve is 15 cases, with the average surgical duration decreasing from 167 to 117 min (*p* = 0.0001) after the first 15 cases. In the TOETVA group, there was a 14.4% overall rate of complications, with no cases of permanent vocal cord weakness or low blood calcium. According to the study, when TOETVA was compared to traditional procedures, there was no significant difference in the rate of complications [29]. This is comparable to studies reporting the learning curve for TOETVA and other endoscopic thyroidectomy approaches [27,28].

In 2021, Chai et al. also presented their experience with TOETVA; however, the results of their analysis differed significantly in terms of the learning curve. The findings of their study suggest that a surgeon who has extensive experience with open and other endoscopic thyroidectomy techniques (such as the BABA) required approximately 57 cases to become proficient in performing TOETVA, which is much more than the previously mentioned studies [31]. Nevertheless, the surgical time taken to perform lobectomy was similar to other reports of initial TOETVA experience, and the complication rates were comparable to acceptable rates following OT. TOETVA has also been successfully performed in obese patients with a body mass index (BMI) greater than 40 without increased complications or longer surgical times. While the magnified view during TOETVA can provide an excellent view of the RLN, there is a concern about the risk of RLN injury during TOETVA compared to OT. However, the overall incidence of RLN injury in TOETVA was similar to that of open thyroidectomy, and the use of intraoperative neuromonitoring (IONM) during TOETVA can reduce the risk of RLN injury [31,79].

There is also a concern about injury to the mental nerve (MN) during TOETVA, which can cause numbness or loss of sensation in the skin supplied by the nerve [74]. This could lead to changes in the way the patient speaks, eats and performs other activities that involve the mouth and lower jaw. In some cases, the MN may regenerate, and the patient may recover some or all of the lost sensation over time. However, in some cases, the injury may be permanent, and the patient may have to learn to adapt to the changes in sensation. It seems that the way to avoid injury to the MN is to use transoral and submental thyroidectomy (TOaST). One of the latest studies, published in 2022, demonstrated that TOaST can not only achieve a good curative effect but also prevent harm to the mental nerve, reduce pain in the patient’s jaw, achieve a good aesthetic effect and simplify the operation for the surgeon [80].

Surgical site infection (SSI) is another potential complication of TOETVA. It is important to avoid SSI because it can lead to additional complications and a longer recovery time for the patient. SSIs can cause the incision site to become red, swollen and tender and may result in the drainage of pus or other fluids. They can also cause fever and chills and increase the patient’s risk of developing sepsis. As such, there is a significant interest in understanding more about the factors that contribute to the development of SSIs and in finding ways to prevent or reduce the risk of these infections. Recent studies on surgical site infections (SSIs) have examined several topics. These include the use of prophylactic antibiotics, wound dressings and other care products, managing blood glucose levels in patients with diabetes and implementing infection prevention protocols during the perioperative period [24,81,82]. Karakas et al. found that TOETVA was a safe and effective surgical technique with a low complication rate and good cosmetic outcomes. Their study also found that the use of prophylactic antibiotics and enoral mucosal disinfection before surgery may be effective in reducing the risk of infection [82]. However, another study reached a different conclusion, showing that the use of prophylactic antibiotics may not be necessary in TOETVA due to the normal flora in the oral cavity acting as a barrier against potential pathogens [83]. This study also found that this procedure may not require prophylactic antibiotics because there is no invasion of non-resident bacterial flora [83]. However, the study had a small sample size and was retrospective in design, so further studies are needed to confirm these findings. Further investigations are required to conclusively estimate the need for antibiotics in TO surgery. It is important for healthcare providers to stay up to date on the latest research on SSIs in order to provide the best possible care to their patients. By understanding the factors that contribute to the development of SSIs and implementing evidence-based prevention strategies, healthcare providers can help reduce the risk of these infections and improve patient outcomes.

There has been increasing interest in comparing the use of TORT and TOETVA in recent years. Current research indicates that both robotic and endoscopic transoral thyroidectomy are safe and effective surgical techniques, with low rates of complications and good cosmetic outcomes. In 2021, Chen et al. conducted a study in which they compared the safety and outcomes of TORT with those of TOETVA [84]. They found that the median operative time was longer for TORT than for TOETVA (308 (284–388) vs. 228 (201–267) min; *p* < 0.001). However, there was no significant difference in blood loss or pain scores between the two groups. The TORT group had a higher rate of central neck lymph node dissection, with 28 out of 53 (52.8%) TORT patients having the procedure compared to 10 out of 53 (18.9%) TOETVA patients. Nevertheless, the number of total and positive lymph nodes did not differ significantly between the two groups when the procedure was performed. The rates of hypoparathyroidism and RLN injury were also not significantly different between the two groups. There were no conversions to open thyroidectomy, mental nerve injuries or surgical site infections in either group. The learning curve for TORT was 25 cases in this study, but no obvious learning curve was observed for TOETVA. This study suggests that both TOETVA and TORT are similarly effective and feasible options for thyroid surgery. There were no significant differences found between the two groups in terms of operative duration, blood loss, pain scores, rates of hypoparathyroidism or recurrent laryngeal nerve injury or incidence of complications. Yet there is still a lack of research comparing robotic and endoscopic transoral thyroidectomy, so further study is needed in this area.

In recent years, the pursuit of the best cosmetic outcome and the absence of scarring has become a major focus of research and scientific publications in the thyroidectomy field [85,86,87,88,89]. As the aesthetic outcomes of elective procedures become increasingly important, TOETVA represents a ground-breaking advancement in endocrine surgery. An eye-tracking study conducted in 2020 compared the distraction caused by scars on the neck after OT to the distraction caused by scars after transoral endoscopic thyroidectomy using TOETVA or transoral endoscopic parathyroidectomy using the vestibular method (TOEPVA) [90]. The study measured the amount of attention that participants paid to scars on the neck and compared the results between the three groups to see if there were any significant differences. Observers viewed facial images of patients, and eye fixations were recorded in real time. Attention was significantly different towards the faces of those who had undergone open-neck surgery vs. TOETVA/TOEPVA (T2 = 43.66; F(32,131) = 14.5389; *p* < 0.0001), with observers attending more to the neck (0.20 s; *p* < 0.0001; 95% CI, 0.13, 0.26 s) and less to the peripheral face (−0.24 s; *p* = 0.0031; 95% CI, −0.39, −0.08 s) of open-neck surgery patients. These differences persisted even months after surgery (T2 = 13.97; F(3451) = 4.6377; *p* = 0.0033). By contrast, fixation patterns for TOETVA/TOEPVA patients were not significantly different from controls (T2 = 5.59, F(31,186) = 1.8602; *p* = 0.1345). These findings suggest that TO neck surgery may be an effective option for patients who are seeking a cosmetic result that does not distract the attention of others. Zhang et al. presented the results of a study that compared postoperative discomfort in patients who underwent two different surgical approaches: TOETVA and conventional OT. The study used a VAS to evaluate pain levels. The results showed that, overall, both groups had mean pain VAS scores of less than 4.0 at all time points. In the first 24 h after surgery, the TOETVA group had lower VAS scores for neck and back pain and difficulty swallowing compared to the OT group. However, the TOETVA group had higher VAS scores for jaw pain in the first 24 h after surgery. There were no significant differences between the two groups in terms of the number of patients who needed postoperative pain medication or the prevalence of persistent pain requiring prolonged treatment [91]. Comparable results have been published in recent years, and many of these showed that TORT may be a suitable operative alternative for patients who do not want surgery to leave scars on their neck [25,26,29,30,76,80].

In conclusion, TOETVA has been shown to have several potential advantages over traditional OT, such as reduced pain and faster recovery. However, TOETVA may have a longer operative duration compared to OT. This may be due to the fact that the surgeon must work through a small incision and may need to use specialised instruments, which can take longer than using traditional surgical techniques. The exact length of the operative time can vary depending on the specific case and the skill of the surgeon, but it is generally longer than for other techniques such as open thyroidectomy [25,92]. It is important to note that while longer operative duration may be a disadvantage of TOETVA, this is not the only factor that should be considered when deciding on the best surgical approach. Other factors such as the potential for reduced scarring, faster recovery and lower rates of complications may also be important to consider. TOETVA is not suitable for all patients, and, in some cases, a combination of transoral and extracervical approaches may be necessary.

### 3.2. Thermal Ablation

A very promising alternative to classical thyroidectomy is thermal ablation, especially for the treatment of benign thyroid lesions. Patients undergoing this procedure suffer fewer complications and have a better cosmetic outcome and shorter hospitalisation, with a comparably good therapeutic effect [93]. The most commonly chosen thermal ablation (TAb) techniques for the treatment of benign thyroid tumours include radiofrequency ablation (RFA), microwave ablation (MWA) and laser thermal ablation (LTA) [94,95]. RFA uses alternating electric current, MWA uses microwave energy, and LTA uses photon emission. All three are percutaneous techniques [94].

Zhang et al. compared the efficacy of ultrasound-guided percutaneous RFA and open thyroidectomy in the treatment of benign thyroid lesions. The study included 266 patients who were divided into two equal groups. Open thyroidectomy was performed in one group and RFA in the other. The RFA power used in this study oscillated between 20 W and 40 W (average 29.06 ± 3.85 W). In both groups, there was no case of recurrence of either local or distant metastases during the follow-up period. One patient in the RFA group and one patient in the open thyroidectomy group experienced transient hypoparathyroidism, and two patients in the RFA group and four patients in the open thyroidectomy group experienced transient voice change. However, the difference was not significant (*p* > 0.05) for both complications. No cases of dysphagia, permanent hoarseness, haematoma or seroma were reported in either group. RFA, on the other hand, proved to be significantly better (*p* < 0.05) than open thyroidectomy in terms of operative duration (27 min vs. 66 min), hospitalisation (3.8 days vs. 6.8 days) and cosmetic result [96].

Bo et al. compared the efficacy of TAb to open and endoscopic thyroidectomy. They used either RFA or MWA (at 30 W or 35 W), depending on the experience of the operator. Of the 505 patients used for the study, 129 underwent TAb. In the TAb patients, the complete response rate was 96.9%, and the technique efficacy was 93.2%. Tumour regrowth was reported in only 1 out of 129 patients. Three patients experienced transient voice change, and five patients experienced transient haematoma. When comparing TAb with open thyroidectomy after propensity score matching, it scored significantly better (*p* < 0.05) in terms of treatment time (33 min vs. 66 min), hospitalisation (1.4 days vs. 4.0 days) and postoperative hypothyroidism (0.8% vs. 25.4%). In addition, patients in the TAb group were more satisfied with the cosmetic outcome and less likely to experience discomfort after surgery. However, the difference in overall satisfaction was not significant. When comparing TAb with endoscopic thyroidectomy after propensity score matching, it scored significantly better (*p* < 0.05) in terms of treatment time (30 min vs. 72 min), hospitalisation (0.8 days vs. 3.6 days) and postoperative hypoparathyroidism (0% vs. 18.6%). The difference in overall satisfaction was not significant [95].

For most studies using TAb, RFA was used as the technique, which seemed to have the best results. Crespo et al. noted that the advantage of RFA over MWA may be due to the lack of specially designed microwave antennas for the ablation of thyroid nodules. In their study of 35 patients, they used a novel MWA system of non-cooled 17 or 18 G antennas and a multiple overlapping ablation technique. The size of tumours was examined before the application of TAb and 1, 3, 6 and 12 months after the procedure. There was a decrease in tumour volume by an average of 32%, 60%, 67% and 74%, respectively. In all patients, disease symptoms resolved after treatment, and patients were satisfied with the cosmetic outcome (*p* < 0.05). There were no cases of major complications during the study, although one patient (3.3%) suffered from transient aphonia, which resolved after three months. MWA also does not cause the heat theft that is present with RFA [97].

In conclusion, TAb seems to be a good alternative to traditional open thyroidectomy. This method has a good outcome and a low number of complications. It also reduces the hospital stay and the procedure duration, which results in lower hospital costs and less risk of hospital-acquired infections. Recent reports have paid particular attention to RFA as a good alternative to traditional open thyroidectomy in the treatment of thyroid nodules, especially in high-risk patients. They indicated that patients suffer fewer complications, hospitalisation time is reduced, and the preservation of partial thyroid tissue reduces the need for postoperative hormone therapy [98]. RFA achieves similarly good results in the treatment of both benign lesions and indeterminate thyroid nodules [99].

### 3.3. Identification and Assessment of Parathyroid Function

#### 3.3.1. Autofluorescence Imaging

Removal of the parathyroid glands (PGs) or damage to their blood supply and consequent temporary hypoparathyroidism is the most common complication after thyroidectomy [100]. The incidence of temporary hypoparathyroidism is 20–30%, and permanent hypoparathyroidism occurs in 1–4% of all patients [101]. The risk of this disease after thyroidectomy depends on the extent of the operation, the experience of the surgeon performing the procedure and the number of PGs identified during the operation [100]. Currently, PGs are identified primarily by eye by the surgeon, without the use of any equipment [100,102]. As described above, the detection of PGs is important in preventing the most common complication of thyroidectomy, and their identification depends significantly on the skill of the operator, so a number of methods have recently been developed to assist surgeons in distinguishing PGs from surrounding tissue.

Autofluorescence is the phenomenon of light emission of a longer wavelength by tissue after absorbing light of a particular, shorter wavelength. PGs exhibit autofluorescence under near-infrared light more strongly than the surrounding soft tissues of the neck, making it possible to identify them [100,103]. The seven papers we reviewed evaluated the ability of near-infrared fluorescence imaging (NIFI) in detecting PGs during thyroidectomy. Kim et al., 2016, determined that the average intensity of PGs’ surface autofluorescence relative to the average intensity of background tissue surface fluorescence (P/B) was 2.76, with a minimum of 1.95 and a maximum of 5.20. Additionally, the time required for the preparation of the equipment for imaging before the operation was 5–10 min, and during the operation, the imaging took less than 1 min, suggesting a good result [104]. In another study by Kim et al., 2017, the authors achieved an average P/B of 4.78 (1.96–12.21), and the time required for equipment preparation was about 10 min before the operation and 2–3 min for imaging during the procedure. In addition, they determined the utility of autofluorescence of PGs in their detection at three stages of surgery: group 1—before the surgeon is able to identify them; group 2—after their identification by the surgeon when they were not detected in group 1; group 3—in the excised specimen when group 1 and 2 were negative. They achieved sensitivity, specificity, positive predictive value (PPV), negative predictive value (NPV) and accuracy (AC), respectively, of 92.75%, 100%, 100%, 16.66% and 92.85% for group 1; 98.55%, 100%, 100%, 50% and 98.57% for group 2; and 100% for all values for group 3 [105]. Differences in P/B between studies by the same author may be due to differences in the number of patients—in the first, there were 8 patients and 16 PGs, and in the second, there were 38 patients and 70 PGs. The use of autofluorescence during thyroidectomy was also studied by Ladurner et al. Of the 41 PGs from 20 patients studied, the authors were able to identify 37 PGs using NIFI, with a sensitivity of 90%. This study investigated the efficacy of detecting PGs that had previously been definitively identified visually by the surgeon, and autofluorescence was only used to confirm the identification of PGs. No autofluorescence was revealed in any of the other thyroid, lymph node or adipose tissue. The researchers determined that the identification of PGs using NIFI increased the operation duration by about 10 min [106]. Takahashi et al. studied the effectiveness of autofluorescence NIFI in identifying PGs in two ways. Using tissues resected during thyroidectomy, including PGs removed for auto-transplantations or incidentally, an experienced surgeon identified 28 tissues as potential PGs and 32 as lymph nodes (LNs); NIFI also identified 15 additional tissues as potential PGs. These 75 tissues were re-evaluated by the surgeon and with NIFI and then sent for evaluation with frozen sections. Sensitivity, specificity, PPV, NPV and AC for the surgeon and for the NIFI were, respectively, 100%, 85%, 85.4%, 100%, 92% and 97.1% and 87.5%, 87.2%, 97.2% and 92%, with the *p*-values always well above 0.05. After the histopathological evaluation of all resected tissue, six PGs previously undetected by either method were detected. Thus, sensitivity for the surgeon and NIFI in detecting PGs directly from the excised tissue was also determined. This amounted to 61% for the surgeon and 82.9% for NIFI (*p* < 0.0479). Based on this, it was determined that NIFI can be an aid to novice surgeons in distinguishing PGs from PG-like tissue and tissue suspected of being PGs, and even to experienced surgeons, in identifying PGs from primary thyroidectomy specimens. After analysing the NIFI errors, it was noted that glandular bleeding or encapsulation with thick tissue could cause inferior NIFI performance [107].

Three of these seven papers evaluated the effect of autofluorescence NIFI not only on the identification of PGs but also on the incidence of postoperative hypocalcaemia (PH) after thyroidectomy [108,109,110]. Dip et al. studied a group of 170 patients referred for thyroidectomy, half of whom had the procedure performed with NIFI. It took 3 to 5 min (median 4 min) to assess the fluorescence of PGs. There was no statistically significant difference in the number of PGs detected intraoperatively in the control group of 3.6 vs. the study group of 3.5 (*p* = 0.32). Similarly, there were no statistically significant differences between the control group and the study group in terms of the incidence of PH (<8.0 mg/dL): 16.5% vs. 8.2% (*p* = 0.103), despite a twofold difference in the results; symptomatic PH: 1.2% vs. 2.4% (*p* = 0.56); long-term calcium replacement: 1.2% vs. 1.2% (*p* = 1.00); postoperative hospitalisation required: 14.1% vs. 7.1% (*p* = 0.14). However, significant differences were observed in mean postoperative serum calcium (8.39 mg/dL vs. 8.65 mg/dL (*p* = 0.009)) and in severe PH (<7.6 mg/dL) (11.8% vs. 1.2% (*p* = 0.005)) for the control and study groups, respectively, using NIFI [108]. Benmiloud et al., in 2017, divided 513 patients undergoing thyroidectomy into four groups. Three groups only used white light to identify PGs: control group 1, control group 2 and the NIFI− group; in the NIFI+ group, NIFI was also used to identify PGs. Thyroidectomy in groups 1 and 2 was performed by one surgeon and in the NIFI− and NIFI+ groups by a different surgeon. A statistically significant difference (*p* < 0.05) was observed in the number of intraoperatively identified PGs between NIFI+ and NIFI− (76.3% vs. 65.7%) and NIFI+ and group 1 (76.3% vs. 62.6%); the occurrence of transient PH (<8.0 mg/dL) between NIFI+ and NIFI− (5.3% vs. 20.9%), NIFI+ and group 1 (5. 3% vs. 16.1%) and NIFI+ and group 2 (5.3% vs. 19.5%); the number of patients who underwent PG autotransplantation in NIFI+ vs. NIFI− (2.1% vs. 15%), NIFI+ vs. group 1 (2.1% vs. 16.7%) and NIFI+ vs. group 2 (2.1% vs. 16.1%); and the number of inadvertent PG resections between NIFI+ and group 1 (1.1% vs. 8%). The percentage of identified PGs was defined as the number of detected PGs divided by the number of theoretically present PGs. However, there were no statistically significant differences between the groups in terms of required treatment (*p* = 0.7839) or the duration of PH (*p* = 0.7923). PGs took between 3 and 5 min to identify, but there was no statistically significant difference between the NIFI+ and NIFI− groups in operating duration [109]. In another study, Benmiloud et al., in 2019, divided 241 patients who underwent thyroidectomy into a control group and a group where NIFI was used to identify PGs during the procedure, the NIFI group. A statistically significant difference (*p* < 0.023) was observed between groups in the intraoperative identification of four PGs between NIFI and control (47.1% vs. 19.2%), two PGs between NIFI and control (16.5% vs. 33.3%) and one PG between NIFI and control (1.7% vs. 15.8%); the occurrence of transient PH between NIFI and control (9.1% vs. 21.7%); the number of patients who underwent PG autotransplantation in NIFI vs. control (3.3% vs. 13.3%); the number of inadvertent PG resections between NIFI and control (2.5% vs. 11.7%); and the duration of the operation between NIFI and control (median 99 min vs. 91 min). However, there were no statistically significant differences between the NIFI and control groups regarding permanent hypocalcaemia (0% vs. 1.7%); parathyroid hormone at postoperative day (POD) 1 (33.2 pg/mL vs. 28.6 pg/mL); or the median duration of hospitalisation (3 vs. 3) [110]. In addition, NIFI helped identify between 61.6% and 68% of finally identified PGs during surgery, before they were identified by the surgeon only, with the naked eye [109,110].

Five papers studied the utility of autofluorescence in detecting PGs during thyroidectomy and parathyroidectomy (PTX) [111,112,113,114]. Kose et al. studied the use of NIFI in 310 patients undergoing thyroidectomy or PTX. NIFI was used to detect PGs during the procedure before and after dissection, including in excised specimens. The identification of PGs was verified visually by an experienced surgeon or, in 550 specimens, by pathological confirmation. Autofluorescence was detected in 947/971 (98%) PGs, of which 228 PGs (23%) were visualised with NIFI before being identified visually by the surgeon. Among pathologically confirmed PGs, sensitivity was 194/197, (98.5%); specificity was 343/35, (97.2%); PPV was 194/204 (95.1%); NPV was 343/346 (99.1%); and AC was 537/550 (97.6%). Intraoperative identification of PGs with NIFI took < 1 min. Mean autofluorescence divided by the background tissues of PGs, thyroid and other tissues was, respectively, 1.73, 1.38 and 1.43 (*p* = 0.003) [111]. Falco et al. measured the intensity of the autofluorescence of PGs during operation for 28 patients undergoing thyroid or parathyroid surgery. The identification of PGs was confirmed by histopathological results or visually confirmed by an experienced surgeon. The mean intensity of the autofluorescence of PGs was 40.6 (14.1–67.1), thyroid glands 31.8 (9.5–54.1) and the background 16.6 (1.2–32). PGs showed higher autofluorescence intensity compared with thyroid glands and the background (*p* < 0.0014) [112]. A similar experiment was conducted by McWade et al. who collected fluorescence measurements from 264 PGs in 137 patients undergoing thyroidectomy or parathyroid surgery. The validation of the identification of PGs was conducted as described above. The range of P/B was 1.2 to 29. The time taken for the identification of the PGs by NIFI was approximately 3–4 min of extra operating time. PGs showed greater autofluorescence intensity than thyroid and surrounding tissues in 97% of cases. A patient’s BMI (*p* = 0.0018), the type of disease (*p* = 0.008), preoperative calcium (*p* = 0.012) and vitamin D (*p* = 0.026) levels showed a significant influence on PG autofluorescence intensity [113]. Kahramangil et al., in an analogous study on 210 patients who underwent thyroidectomy and PTX, investigated the utility of NIFI for the detection of PGs. The identification of PGs was confirmed by the histopathological result or visually confirmed by an experienced surgeon. In total, 594 PGs were identified, 584 of which were also identified with NIFI, with a sensitivity of 98%. In addition, 46% of PGs were identified with NIFI before the surgeon was able to identify them by sight. The identification of one PG with NIFI took about 1 min intraoperatively [114]. A meta-analysis summarised the results of seventeen papers on 1198 patients. It investigated the utility of autofluorescence NIFI for detecting PGs during thyroidectomy and PTX. PGs in the cited papers were confirmed visually or by histological validation. The summary of the statistical analysis showed a sensitivity of 96.93%, specificity of 92.48%, PPV of 94.88 and NPV of 95.17 in identifying PGs using autofluorescence NIFI [115].

Autofluorescence appears to be a useful tool during thyroidectomy. The aforementioned studies have shown that the use of autofluorescence NIFI is a process that does not significantly prolong surgery time. In addition, PGs show significantly higher autofluorescence than surrounding tissues, with a P/B of 1.2–67.1. Autofluorescence also appears to increase the number of PGs detected intraoperatively, increase the number of PGs autotransplanted and decrease the number of PGs accidentally removed. Additionally, it appears to reduce the number of transient PH and severe PH. However, none of the studies noted any effect of autofluorescence NIFI on permanent hypocalcaemia or the number of patients requiring prolonged treatment or prolonged hospitalisation. Taking all of the examined papers into account, however, it seems that autofluorescence NIFI can be a helpful tool during thyroidectomy to preserve PG function for both novice and experienced surgeons. A summary of the papers and results describing the effectiveness of autofluorescence NIFI in identifying PGs can be found in Table 2.

#### 3.3.2. Indocyanine Green Fluorescence (ICGF)

Zaidi et al. studied the utility of ICGF in identifying and evaluating PGs during thyroidectomy in 27 patients. Fluorescence in PGs was visible approximately 1 min after injection and persisted for up to 20 min. Of the 85 PGs identified by the surgeon, 71 (84%) showed ICGF. In addition, four types of PG fluorescence were determined depending on the number of PGs that showed ICGF: 0—no uptake, 1: <30%, 2: 30–70% and 3: >70%. A statistically significant difference in mean PTH level in POD 1 was observed between patients with at least two PGs marked as type 1 (PTH = 9 pg/dL) compared to patients with fewer than two PGs marked as type 1 (PTH = 19.5 pg/dL; *p* = 0.05) [116]. Yu et al. studied the effect of ICGF use on the bilateral axillo-breast approach robotic thyroidectomy outcome in 66 patients, in 22 of whom ICGF imaging of PGs was used during surgery. Transient hypoparathyroidism was defined as a decrease in serum PTH below 15 pg/mL, and permanent hypoparathyroidism was defined as a decrease in the serum PTH below 15 pg/mL with the need for oral calcium supplementation for longer than 1 year. The average time between indocyanine green injection and PG fluorescence was 203 s (125–331 s), which lasted an average of 20.8 min (16.6–35.8 min). Only inferior PGs were identified and, of those targeted with ICGF, all of them were visualised. There was no difference in the incidence of postoperative transient or permanent hypoparathyroidism between the groups. The only statistically significant difference was the amount of incidental PTX. This occurred in seven patients (15.9%) in the control group and none of the patients in the group where ICGF was used to identify PGs (*p* = 0.048) [117]. Van den Bos et al. studied the subjective and objective utility of ICGF in identifying PGs during thyroidectomy in 30 procedures for 26 patients. In their calculation, they defined one operation as one separate patient; hence, we treated their study as including 30 patients, as intended by the authors. The average time required for ICGF testing was 5 min and 35 s. Among the 30 patients, 41 PGs were identified in 25 patients using vision alone, and 31 PGs were identified in 23 patients using ICGF. In three patients, no PGs were identified, and, in two patients, PGs were identified only by using ICGF. In 17 (57%) patients, ICGF was determined by surgeons to be useful in identifying PGs. Among the remaining 13 patients, in 6 of them, PGs were clearly identified visually, and no confirmation was needed; in 3 of them, PGs remained black after indocyanine green injection; and in 4 of them, the surrounding PGs were too fluorescent to distinguish PGs from the surrounding tissue [118]. Rudin et al. studied the effect of ICGF use during thyroidectomy on the assessment of PG function after surgery. The study included 210 patients, in 86 of whom, ICGF was used to assess PGs. Among these 86 patients, 281/344 or 82% of PGs were detected. In 87% of the ICGF group, the visual assessment of blood supply to PGs was in agreement with an assessment using ICGF. Of the remaining 13%, 6% of patients had a good blood supply of PGs visually and none with ICGF, resulting in the autotransplantation of these PGs, while 6.8% of patients had a present blood supply visualised with ICGF and no blood supply visible only by a visual assessment. In POD 1, PTH was <15 pg/mL in 36% of patients in the control group and in 37% of patients in the ICGF group. The mean PTH level in POD 1 was 21 pg/mL in the control group and 19 pg/mL in the ICGF group. PTH of <6 pg/mL was present in 14% of patients in the control group and in 15% of the ICGF group. Autotransplantation was significantly more frequently performed (*p* = 0.0001) in patients in the ICGF group than in the control group (36% vs. 12%). It was noted that the presence of two or more well-circumscribed PGs visible with ICGF was associated with postoperative PTH levels of >15 pg/mL (*p* = 0.044). The AC, sensitivity and specificity of the presence of two or more PGs in predicting PTH > 15 pg/mL were 63%, 72% and 50%, respectively [119]. Zaidi et al., in their other experiment, studied the utility of ICGF in identifying PGs during PTX in 33 patients. Fluorescence in PGs was visible approximately 30–60 s after injection and persisted for up to 20 min. Of the 112 PGs identified by the surgeon, 104 (92.9%) showed ICGF. In addition, four types of PGs fluorescence were determined, which was similar to the previous study, depending on the percentage of parathyroid volume that showed ICGF 0: no uptake, 1: <30%, 2: 30–70% and 3: >70%. The researchers showed no effect of gender, BMI, PTH or vitamin D on ICGF uptake. However, a higher degree of fluorescence was shown in patients with preoperative calcium of >11 mg/dL (*p* = 0.04) and when PGs were greater than 10 mm (*p* < 0.01) [120].

This work suggests the utility of ICGF during thyroidectomy, primarily in predicting the postoperative functionality of PGs. The identification of PGs in the papers discussed was between 80 and 90% and, thus, worse than in the autofluorescence studies. However, we note the utility of ICGF in assessing PTH levels postoperatively. The finding of fluorescence in two or more PGs was usually indicative of higher postoperative PTH values in patients. The utility of ICGF in thyroidectomy and parathyroidectomy is shown in Table 3.

#### 3.3.3. Others

There are other methods for identifying PGs or assessing vasculature, such as using fluorescence with fluorescent methylene blue, 5-aminolevulinic acid, coherence tomography or laser speckle contrast imaging. However, these techniques have many side effects or have not yet been properly studied. Methylene blue has shown numerous side effects including neurotoxicity. After the administration of 5-aminolevulinic acid, patients must be protected from sunlight for 24–48 h due to its phototoxicity. Coherence tomography has been the subject of a small number of studies, mainly ex vivo. Laser speckle contrast imaging shows very high sensitivity to movements including the patient’s breathing, making it difficult to obtain reliable results. For these reasons, we did not describe these studies in detail, as there are not yet enough reliable studies indicating the growing potential of these methods [100].

### 3.4. Perioperative Bleeding

#### 3.4.1. Introduction

The vascularisation of the thyroid gland consists of the symmetrical superior and inferior thyroid arteries, which are branches of the thyro–cervical trunk of the subclavian artery. Occasionally, the inferior thyroid artery is present, which may be a branch of the brachiocephalic trunk of the internal thoracic artery or the aortic arch. These arteries are connected by intrathoracic anastomoses to form the disc network. The veins produce the extrathoracic thyroid plexus and the odd thyroid plexus at the level of the smokestack. Blood drains through the superior and middle thyroid veins into the internal jugular veins and also through the inferior thyroid veins, which flow with a common trunk into the brachiocephalic vein. Blood flow through the thyroid gland is high, at 2500 mL/kg tissue min ×min, compared to 180 mL/kg tissue min ×min through the lungs [121,122].

Postoperative bleeding is one of the most common complications following thyroid surgery. The incidence rate varies between 0.1% and 4.2%. However, in highly specialised centres, the rate does not exceed 1% of all operations. Haemorrhages originate from the subcutaneous tissue, subglenoid muscles, upper pole, common jugular vein, tissue adjacent to the retrograde laryngeal nerve, thyroid stump and trachea [122,123,124,125,126,127,128,129].

Bleeding can be divided into two categories: deep haemorrhages and shallow haemorrhages. The main symptom of shallow bleeds is a haematoma on the skin. The symptoms that accompany deep bleeds are mainly a feeling of tightness in the neck area, coughing, shortness of breath, stridor, tachycardia, swallowing problems, change in voice, feeling cold/warm and anxiety; additionally, it must be remembered that in this location of bleeding, there are no changes visible on the skin. In terms of the risk of death, deep bleeding is significantly more serious. In addition, the undifferentiated symptoms of deep bleeding are an enlargement of the neck circumference, dysphagia and an increase in the volume of secretions drained by 150 mL. Additionally, it has been noted that patients experiencing postoperative bleeding report more than one symptom [122,123,129].

#### 3.4.2. Risk Groups

It is both simple and good practice to classify patients into low-risk and high-risk groups before the planned operation in order to avoid bleeding. In this way, possible complications can be prepared for in advance, and special care can be taken during the procedure itself. We distinguish between preoperative, intraoperative and postoperative risk factors for bleeding.

Preoperative risk factors include male sex (*p* < 0.00001), older age (*p* < 0.00001), Graves’ disease (*p* < 0.00001), ≥3 metabolic diseases, obesity, retrosternal goitre (*p* = 0.003), previous surgery on the thyroid gland (*p* = 0.01) and haemorrhagic diathesis. Specifically, a case report from the Nippon Medical School in Tokyo described haemophilia A, which showed values of the coagulation determinants PT or aPTT in the normal range. This demonstrates the need to ask during the taking of a history and make a note of any family members with chronic coagulation diseases [123,124,127,128,129,130,131,132,133]. The risk factors are combined and shown in Table 4.

The issue of intraoperative risk factors was addressed in a study by Edafe et al. Patients who underwent total thyroidectomy were more likely to require reoperation for bleeding compared with hemithyroidectomy (*p* = 0.045) or parathyroidectomy (*p* = 0.001). According to other authors, the duration of surgery and the monthly number of operations performed by a given surgeon are also important. According to some studies, a surgeon completing more than four thyroid operations per month is an indicator of high volume, and this is associated with statistically less bleeding as a complication after thyroidectomy [124,127].

Postoperative risk factors include vomiting, increased blood pressure, postoperative pain and anaemia [122,134].

#### 3.4.3. Operating Techniques

As studies show, it is possible to reduce the risk of bleeding during procedures by using several methods. In patients undergoing open thyroidectomy, the importance of ligating blood vessels and placing clips is emphasised. In addition, the Valsalva manoeuvre should be used before closing the surgical field, due to the fact that the patient’s pressure during general anaesthesia is lower than under normal conditions. Raising the pressure helps visualise bleeding that could not be seen at a lower pressure [122,135]. A 2020 study from Istanbul University suggests that it is useful to apply a peak airway pressure (PAP) of 50 cm H_2_O for 22.5 s to help visualise bleeding sites. This is an extended study including different pressures and times, which updates the first study on this topic from 2015 [136].

Interestingly, there is no published evidence to support the commonly used technique of fitting drains to a wide range of patients to prevent bleeding. It does not reduce the incidence of bleeding, haematoma or reoperation but, interestingly, increases the length of hospitalisation. Therefore, it is worth reconsidering the need to insert drains in low-risk patients. In contrast, it is worthwhile to place them in patients at higher risk [126,133].

The ITSRED FRED protocol, which has existed for 4 years, can be used when performing surgery to avoid bleeding. The name of the protocol is an acronym for the specific steps during surgery that need to be carried out prior to wound closure to minimise the risk of bleeding. These include flushing with water to remove blood clots and identify other areas of bleeding, pausing to check for bleeding, raising the systolic pressure to more than 100 mm Hg, relieving pressure in the neck, raising the venous pressure (via the Valsalva manoeuvre, the head-down position or both), applying aids and assessing bleeding under the flaps [127]. Table 5 shows the acronym ITSRED FRED with an expansion of the content.

The use of Tachosil^®^ haemostatic patches appears to reduce postoperative bleeding, although no statistical significance has been demonstrated in this regard. The best results were achieved in a group of patients treated with antiplatelet agents and/or anticoagulants, whereas, for a small study group, no greater efficacy was demonstrated than by using standard techniques to prevent vascular bleeding, such as vessel protection with the BiClamp^®^, LigaSure^®^ or Harmonic Scalpel^®^ system [130,137].

An interesting solution for the early detection of bleeding is the use of a special trocar with a transparent head (first available in 2018). This allows for the real-time observation of the surgical field and the depth of the surgical instruments. It is also possible to observe and detect areas of bleeding when removing the trocar at the end of surgery. This trocar has been used by surgeons at the Second Affiliated Hospital Zhejiang University School of Medicine in total endoscopic thyroidectomy (TET) operations [138].

#### 3.4.4. Postoperative Management

Bleeding is most common up to 8 h after surgery, and approximately 85–95% of bleeding occurs during the 24 h after surgery. Therefore, at least 24 h of postoperative observation is an important part of care, as medical staff can respond effectively. Careful postoperative observation is crucial for preventing bleeding complications. An important factor is trained medical staff who are aware of the, often uncharacteristic, symptoms of bleeding. This is important because a rapid response is crucial, as bleeding can lead to death. To this end, a specific protocol for the management of postoperative bleeding symptoms—SCOOP—was developed four years ago. Its key is the rapid response of staff during postoperative follow-up [122,123,124,125,126,127,133,134,137,139]. The protocol starts with noticing and identifying symptoms that may indicate bleeding, such as increased swelling of the surgical wound and gas exchange disturbances including tachypnoea, stridor and low saturation. Management involves early oxygen therapy initiation, intravenous hydrocortisone administration and immediate staff involvement. If intubation is not possible, the airway should be secured by tracheotomy or conicotomy [127].

An important element of postoperative care is increased attention for patients with chronic use of antihyperglycaemic drugs (*p* < 0.05) and severe postoperative pain expressed by high values on the NRS pain scale (*p* < 0.001). The use of ketorolac, a drug belonging to the NSAIDs with antiplatelet properties, in the treatment of postoperative pain is an intrinsic risk for haemorrhage (*p* = 0.032) and should, therefore, be avoided [139].

#### 3.4.5. Others

Outpatient thyroidectomies are a dangerous option that may result in worse and prolonged patient protection in terms of postoperative complications, especially in the context of bleeding. It is, therefore, advisable, especially in situations of increased risk of postoperative bleeding, to always admit patients to the hospital [122].

### 3.5. Recurrent Laryngeal Nerve

#### 3.5.1. Introduction

The recurrent laryngeal nerve is a branch of the vagus nerve. It contains motor, sensory and parasympathetic fibres. The nerve is divided into an external and an internal branch. The internal branch is responsible for the sensory innervation of the laryngeal region, while the external branch is responsible for motor innervation. The nerve supplies the four internal muscles of the larynx: the thyroarytenoid, the lateral and posterior cricoarytenoid and the transverse and oblique arytenoid [140]. The RLN branches off in the upper thoracic region, then turns around and ascends in the neck, but the exact courses of the left and right nerves are different. The left hooks into the aortic arch and is longer, while the right turns back at the first part of the subclavian artery. On both sides, the nerves enter the larynx at the site of the cricothyroid joint [140,141]. There are some anatomical variations that affect the incidence of RLN injury during thyroidectomy. Non-RLN is a variation in which the nerve enters the larynx directly, without first descending into the upper thoracic region. This occurs with a frequency of 0.2–1.5% and 0.04% on the right and left sides, respectively, and is a risk factor for paralysis [141,142]. In contrast, extralaryngeal branching of the RLN is more common, affecting 30–78% of cases. Branched RLNs are twice as likely to be damaged as non-branched ones (15.8% vs. 8%) [143]. The gold standard to prevent RLN paralysis is its early identification [144]. Depending on the circumstances, different surgical access is used to identify the nerve. These can be divided into lateral (most common), inferior, superior and medial [145].

RLN palsy is a relatively common complication of thyroidectomy. There are several mechanisms of injury: transection, clamping, ligation, compression, traction, thermal injury and ischaemia. The most common are traction (52.6%) and compression (38.8%) injuries [146]. The main symptom of these injuries is hoarseness, but bilateral injury can also lead to severe ventilatory dysfunction [147]. Therefore, it is important to develop techniques to reduce the incidence of this complication and its treatment.

#### 3.5.2. Intraoperative Neuromonitoring—Prevention of RLN Paralysis

Intraoperative neuromonitoring (IONM) is a technique for monitoring RLN function by recording the electromyographic signal from the vocal muscle. It assists in localising the RLN, and there are publications reporting that its use results in a reduction in the rate of nerve paralysis during thyroidectomy [148,149]. Therefore, during thyroidectomy, IONM is commonly regarded as an adjunct to the gold standard of the visual identification of the nerve. The mechanism of action involves recording the depolarisation of the vocal muscle by recording electrodes, after its stimulation by stimulating electrodes. Monitoring endotracheal tube (ETT) electrodes are used, which are placed at glottal level and are in direct contact with the surface of the vocal cords. Alternatively, it is also possible to use standard endotracheal tubes together with adhesive electrodes. Stimulation is possible through the use of a sterile probe [150]. There are also techniques for percutaneous stimulation [151]. Although the use of IONM has been standardised since 2011, the conduct of neuromonitoring in different centres shows low homogeneity. International guidelines formulate standards for aspects such as equipment configuration, endotracheal tube placement, anaesthesia and the assessment of signal loss. However, the issue of possible surgical access to identify the RLN while conducting neuromonitoring has not been mentioned [150]. Thus, there is still ample room for surgical development in this direction. It is hoped that future studies will isolate the most optimal approaches.

The standard is intermittent IONM (IIONM), during which stimulation is performed at intervals. This runs the risk of delaying the registration of nerve irritation. Continuous IONM (CIONM) is a technique that allows real-time monitoring. As a result, the surgeon is warned of possible RLN injury earlier during the procedure, so its risk is reduced [152]. Schneider et al. showed that CIONM is associated with a lower rate of permanent vocal cord palsy than IIONM [153]. However, a study by Zhao et al. in an animal model revealed some limitations of CIONM. With endoscopic surgery, CIONM with percutaneous stimulation seems not to be possible [151]. However, the work of Chen et al. shows the feasibility of CIONM in the transoral endoscopic thyroidectomy vestibular approach (TOETVA). In this technique, the stimulating electrode was placed directly on the vagus nerve. The study was conducted on 20 patients undergoing TOETVA, and all procedures were successfully completed [154]. The concept of using continuous RLN monitoring with a minimally invasive technique is attractive, so we do not rule out that further studies in the future will enable its implementation.

An ETT-based monitoring system is not perfect. The main difficulty is maintaining constant contact between the electrodes and the vocal cords [155]. In the vast majority of cases, signal reception dysfunction is caused by the displacement of the ETT electrode [150]. The cause of a false loss of signal (LOS) can also be the accumulation of saliva at the site of the recording electrodes. Other limitations include the relatively high cost of the tube, the need for proper sizing and the need to involve an anaesthesiologist to verify its position [155,156]. As proper ETT placement is difficult, Kriege et al. compared direct laryngoscopy with videolaryngoscopy. They used the C-MAC^®^ videolaryngoscope (Karl Storz^®^, Tuttlingen, Germany). The incidence of an inadequate EMG signal after initial ETT placement was 27% and 9% for the direct laryngoscopy group and C-MAC group, respectively (*p* < 0.001). This result suggests that the choice of technique for ETT insertion may affect signal quality during IONM. The videolaryngoscope gives better results, so it would be reasonable to use it as the first-choice device [157].

Knowing the limitations of ETT electrodes, new and better solutions are being sought. There are attempts to use more invasive methods of monitoring RLN during thyroidectomy, via needle recording electrodes. Zhao et al. conducted a study on a porcine model, in which they evaluated and compared ETT needle electrodes placed on the thyroid cartilage (TC). They compared nine parts of the TC and different needle insertion depths to find the optimal electrode site. They showed that guiding the needle into the avascular part of the TC, perichondrally, was safe. The recorded EMG parameters unquestionably reveal the superiority of TC and ETT electrodes. Both types of electrodes achieved comparable latencies; however, the amplitude response to RLN damage was faster with TC electrodes. In addition, EMG signals were more stable during tracheal manipulation, and amplitudes were higher. The lower cost of these electrodes is also one of the advantages [158].

A study by Chiang et al. specifically looked at patients undergoing thyroidectomy. They evaluated the effectiveness of needle TC electrodes placed into the sub-perichondrium. Both ETT and TC electrodes were used in each subject, and their signals were compared. As in the Zhao et al. study, the latencies were comparable, while the amplitudes recorded by the TC electrodes were higher and more stable. However, the authors also reiterated the limitations of the described method, which are that it requires TC exposure, so it is a highly invasive procedure, and, in addition, needle insertion may be difficult in older patients due to a high degree of cartilage calcification [159].

Adhesive electrodes are a less invasive alternative. Wu et al. conducted a study on a porcine model in which they evaluated transcartilage surface electrodes for the recording of laryngeal EMG signals. This was shown to be an effective method. Moreover, transcartilage surface electrodes showed less variation during experimental tracheal displacement than when using ETT electrodes. Other advantages of such adhesive electrodes are their ease and speed of placement and decreased invasiveness while causing less trauma to laryngeal tissues. These electrodes are also relatively inexpensive. However, their limitations should be noted. Their use required a larger skin incision to expose the TC, and the average amplitudes achieved were lower compared to those recorded with ETT electrodes. Therefore, currently, this is not a technique that actually reduces the invasiveness of the procedure and is not an ideal alternative to ETT. Further research into the use of transcartilage surface recording electrodes is required. It is possible that in the future they can be improved and introduced into daily use [155]. Another type of adhesive electrode is electrodes stuck to the skin. Lee et al. conducted a study on a group of 30 patients undergoing thyroidectomy. Each underwent IONM using two types of electrodes: ETT and adhesive skin electrodes. A total of 39 nerves at risk were monitored. The use of skin adhesive electrodes was successful in all cases, so the utility of adhesive electrodes was confirmed. This method reduced the incidence of false LOS and lowered the cost of the IONM procedure. In addition, the authors noted the possibility of using this type of electrode in paediatric patients, where the size of the monitoring tube makes the use of ETT electrodes much more difficult. The main disadvantage of skin adhesive electrodes, however, is the lower obtainable amplitudes of EMG signals. Despite this limitation, the described method demonstrated efficacy, so its routine introduction as an alternative to currently used techniques may be beneficial for patients undergoing thyroidectomy [156].

As the frequent occurrence of LOS due to ETT displacement during IONM is the main problem, the introduction of other types of electrodes could improve the quality of RLN monitoring.

#### 3.5.3. Surgery Technique—Prevention of RLN Paralysis

Zhang et al. investigated the relationship between the choice of surgical technique and the risk of RLN injury during thyroidectomy. They compared the incidence of nerve injury during the open thyroidectomy approach (OTA) and the endoscopic thyroidectomy via bilateral areola approach (ETBAA). The study included 1420 nerves at risk. Patients were divided into three groups: those who underwent ETBAA; those who had contraindications to ETBAA, so they underwent OTA (OTA-H); and a group of patients who had no contraindications to ETBAA but chose OTA (OTA-L). The percentage of patients who experienced temporary postoperative vocal cord palsy was 7%, 2% and 4% for the ETBAA, OTA-L and OTA-H groups, respectively. The rate of RLN injury was, therefore, significantly higher in the ETBAA group. The difference between the ETBAA and OTA-L groups was statistically significant (*p* < 0.0125). ETBAA is a more modern technique with numerous advantages. These primarily include better cosmetic results as well as better visualisation of the superior laryngeal nerves and some lymph nodes. However, in the context of the prevention of RLN damage during thyroidectomy, the classical OTA is a safer method of surgery [160].

#### 3.5.4. Artificial Intelligence (AI) Techniques—Prevention of RLN Paralysis

Gong et al. created a deep learning (DL) algorithm that identifies and measures the width of RLNs in intraoperative images taken freehand by surgeons. The image dataset consisted of 277 images acquired from 130 patients undergoing total thyroidectomy or lobectomy. The performance of the algorithm was investigated, and the effect of image capture conditions on its performance was evaluated. It was shown that, under optimal conditions, the algorithm successfully identified and estimated the width of RLNs in the images. Conditions under which the highest quality images were achieved were close-up distance and medium lighting. The described DL algorithm could be an aid to intraoperative decision making. The use of DL techniques that effectively identify the RLN could reduce the incidence of its damage during a thyroidectomy procedure. However, the described AI system has some limitations. Currently, it is not perfect and does not always accurately find the boundaries of the RLN, especially in the case of nerves that are smaller in diameter, irregular in shape or poorly exposed in the image. In addition, the algorithm’s performance is highly dependent on the conditions in which the images are taken. Establishing standardised parameters for capture distance and lighting is a logical way forward to improve this method. Further research on the use of DL techniques to identify RLNs during thyroid resection surgery is, therefore, required [161].

Augmented reality (AR) is a modern technique that allows virtual images of organs to be superimposed on real structures during surgery. It is used in robotic surgery; however, it is not popular. This is because, at present, it is a cumbersome method to use, as AR images do not move with the movement of the robot and must constantly be manually adjusted. Lee et al. used a vision-based tracking system to enable AR images to be coordinated with camera movements during the procedure. AR images of the common carotid artery, trachea and RLN were constructed based on CT images, and this was tested on nine patients undergoing thyroid surgery. No RLN paralysis complication was reported, suggesting that this type of tracking system could be a significant help to the surgeon when performing robotic thyroidectomy. Robotic surgery is associated with leaving a smaller scar, but the doctor’s inability to use his or her sense of touch and the difficulty in identifying hidden anatomical structures have slowed its rise in popularity. The solution proposed by Lee et al. can help to identify certain important structures and overcome this limitation. Better awareness of the location and anatomy of the RNL during robotic surgery would reduce the incidence of injury to this nerve [162].

As mentioned above, the presence of non-RLN is a risk factor for paralysis [141]. Zhang et al. investigated the feasibility of robotic thyroidectomy in non-RLN patients. The procedure was performed in two patients with right non-RLN, using a bilateral axillo-breast approach. The published paper includes information on techniques for monitoring, isolating and protecting nerves using robotic surgical equipment. In addition, the authors provide intraoperative videos in which these techniques are demonstrated and discussed. In both cases, the nerves were dissected correctly, and the procedure proceeded without damage. These are the first such documented cases in which the procedure was performed using a robot and was successful, i.e., there was no iatrogenic non-RLN injury. The most important advantages of using robots in surgery are highlighted. They provide better precision in tissue manipulation and increased visualisation of the narrow surgical area. This suggests that robotic surgery may have potential to prevent the occurrence of complications of thyroidectomy [142]. Some anatomical variations increase the risk of vocal cord paralysis, including non-RLN. The increased precision associated with the use of a surgical robot may be helpful in preventing the occurrence of this complication, including in cases of atypical nerve anatomy. Further research on the topic is advisable, as it could have significant benefits for the development of thyroid surgery and for patient prognosis.

RLN palsy is a complication that every surgeon fears when performing a thyroidectomy procedure, which can result in significant discomfort. Therefore, it is important to make sure it does not occur. Currently, there are techniques that reduce the risk of damage, but they are far from perfect. Recent research on improving the quality of IONM by using alternative types of monitoring electrodes seems to be a good way forward. Adhesive electrodes are more resistant to tracheal movements and less likely to be displaced compared to ETT electrodes. This results in more stable EMG signals, with a lower rate of false LOS. AI techniques can be a powerful tool to help identify critical neck structures, which include the RLN. Improved nerve identification performance is associated with reduced nerve damage during surgery. Robotic thyroid surgery demonstrates improved precision, which is crucial in terms of preventing RLN damage. Conducting further research to overcome the current limitations of robotic surgery is, therefore, advisable. Introducing such techniques into routine use in the clinic could greatly benefit patients and reduce the incidence of thyroidectomy complications.

#### 3.5.5. Treatment of RLN Paralysis

The return of vocal and swallowing function occurs after the postoperative unilateral interruption of the retrobulbar laryngeal nerve. There are several possible ways in which this may happen. There is a chance of laryngeal muscle involvement by nerves such as the superior laryngeal nerve or the inferior laryngeal nerve lying symmetrically on the other side (via intrathoracic connections). There is also bilateral innervation of the laryngeal muscles. This spontaneous reaction of restoring the motor nerve connection can occur in two ways: “favourable synkinesis” or “unfavourable synkinesis”. Favourable synkinesis occurs when the muscle tone in the paralysed vocal fold is preserved or returns to normal, such that the adjacent vocal fold is able to come into contact with little or no airflow. Unfavourable synkinesis occurs when there is an inferior vocal outcome or, rarely, laryngeal spasm [163].

During surgery, if the IONM signal is lost, nerve injury can be confirmed by ruling out problems with the apparatus using the troubleshooting algorithms from the International Standards Guideline Statement by Randolph et al. If possible intraoperatively, the RNL injury should be categorised as transected or non-transected. If the injury is non-transverse, and the RNLs do not improve during surgery, a wait-and-watch (W&W) technique can be used. In addition, studies by Rosen et al. and Sridharan et al. suggest that the administration of calcium channel blockers, in this case, nimodipine, may be helpful in recovery. This was subsequently replicated with good results in a study by Mattsson et al. On the other hand, if the mechanism of injury was transection, immediate intraoperative repair of the RLN (IIORRLN) may be performed [145,150,163,164,165,166].

The direct connection of the two cut ends of the RNL, RLN to ansa cervicalis, RLN to the vagus nerve and free nerve graft—is used. If possible, an RLN-to-RLN connection is the most preferred option. This should ideally be performed during the primary operation. Microsurgery, bonding or grafting can be used for the fusion. The microsurgery technique is based on the use of non-dissolvable microsutures between 7.0 and 9.0. They are placed so that the two nerve ends are connected to each other using four to six sutures. This is recommended when the defect is no more than 5 mm, and the connection can be made without stretching. If these conditions cannot be met, grafts are recommended. Fibrin glue and cyanoacrylate glue are used for bonding, although they are not popular due to their high rates of adverse effects. The greater auricular nerve, the transverse cervical nerve, the supraclavicular nerve or the supraclavicular nerve can be used in transplantation. According to a study from the Kumamoto University School of Medicine, grafts performed using the great auricular nerve and the ansa cervicalis nerve had the best results, with both providing long-term good postoperative function [163,166,167,168,169].

The use of a laser may, in the future, also be a possible technique for nerve anastomosis. A study by Bhatt et al. compared the use of a KTP (potassium titanyl phosphate) laser and microsurgery. In an animal model, better results were shown in the recovery of vocal function within 6 months following laser anastomosis [170].

## 4. Conclusions

In recent years, there have been many reports of new surgical techniques that seem to match the traditional open thyroidectomy in their efficacy, do not cause significantly more perioperative complications and may even achieve better results in some respects. Major new trends in recent years include new approaches such as TAA, BABA and TOETVA. They are particularly distinguished from traditional procedures by better cosmetic effect (GTAA, TOETVA), reduced hospitalisation time (GTAA, GETTA, BABA), a shorter operation duration (GETTA, BABA–RT) and less postoperative pain (GTAA, GETTA).

A particularly noteworthy innovation is thermal ablation. It stands out from the other options because it is a percutaneous technique, providing a very good cosmetic effect and significantly reducing the duration of the procedure itself and the patient’s hospitalisation. Its final result is comparable to that of other techniques, and the number of serious complications is small.

Another important innovation in recent years is methods of intraoperative parathyroid monitoring using autofluorescence and ICGF. Autofluorescence does not prolong the duration of surgery and significantly minimises the risk of accidental parathyroid resection. ICGF, on the other hand, is proving to be a good means of assessing postoperative PG function.

The management of perioperative haemorrhage control and RLN monitoring has not changed very much in recent years, but a lot of work has been performed on evaluating existing procedures and improving them somewhat, so this is what we have mainly emphasised in our work. Here, the importance of new types of surgical access and, in the future, the use of artificial intelligence algorithms is emphasised.

## Figures and Tables

**Figure 1 cancers-15-02931-f001:**
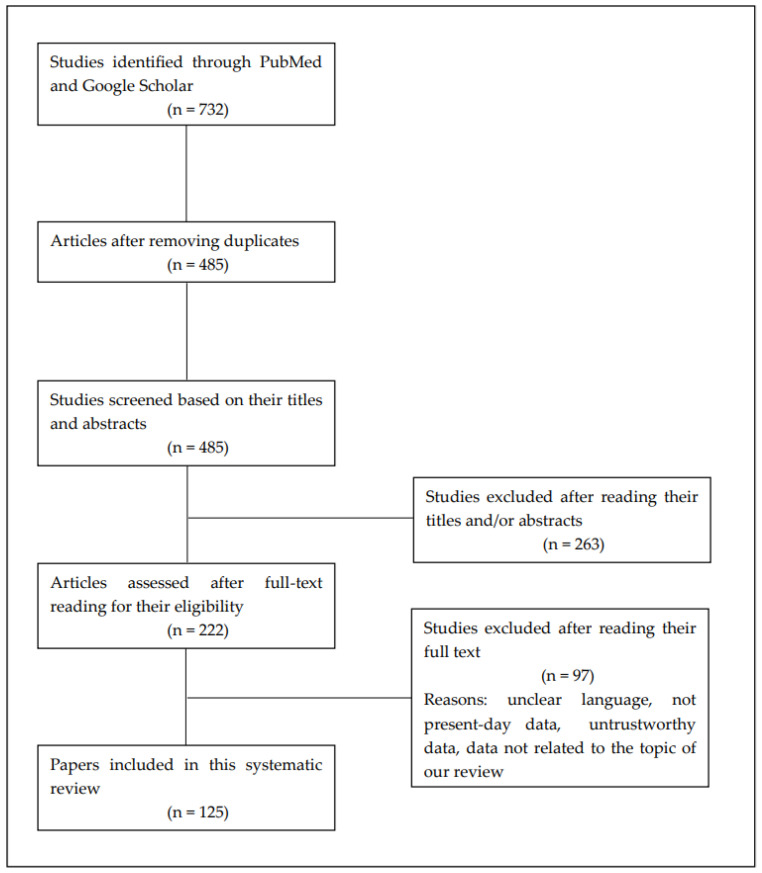
The process and criteria for inclusion of papers in our systematic review.

**Table 1 cancers-15-02931-t001:** A comparison of different surgical techniques used in thyroidectomy.

Approach	Indications	Contraindications	Advantages	Disadvantages	Main Complications	Experience Required	Potential Advancements
Conventional Open Approach	Large * or complicated thyroid tumours, previous neck surgery	N/A	Wide surgical field, easy access to entire thyroid gland	Visible scar, higher risk of complication	Haematoma, infection, vocal cord paralysis, hypoparathyroidism	Any **	N/A
TAA	Small to large thyroid tumours, good neck mobility	Previous neck surgery, goitre, large thyroid tumours	Minimally invasive, hidden scar, faster recovery	Limited access to entire thyroid gland, longer operation time and higher total treatment costs compared to open surgery	Haematoma/seroma formation, hypoparathyroidism, paraesthesia, infection	Intermediate	Improved robotics, transitioning to GTAA for enhanced experience in oncological thyroidectomy
BABA	Small to large thyroid tumours, good neck mobility	Previous neck surgery, goitre	Symmetrical view of thyroid, use of same methods as open surgery possible	Wider surgical flap, higher risk of pain and sensory impairment compared to other minimally invasive techniques	Haematoma, infection, vocal cord paralysis, hypoparathyroidism	Intermediate	Development of mini-flap BABA, improvements in robotics
TOETVA	Small to medium thyroid tumours, good neck mobility	Large thyroid tumours, previous neck surgery, goitre	Minimally invasive, no scar, faster recovery	Limited access to entire thyroid gland, high learning curve for surgeons, small working space, potential for postoperative seroma and infection	Haematoma, infection, vocal cord paralysis, hypoparathyroidism	Advanced	Further research and development of the technique

N/A—not applicable, GTAA—gasless trans-axillary approach. * When referring to the size of a thyroid tumour, “small” typically means less than 2 cm in diameter, “medium” means 2–4 cm in diameter, and “large” means greater than 4 cm in diameter. These sizes are approximate and may vary depending on the specific criteria used by different surgeons or medical centres. ** When referring to the required experience, “any” means that the approach can be performed by surgeons with any level of experience, including those who are just starting out in their career. “Intermediate” means that the approach may require a moderate level of experience and skill and is typically suitable for surgeons who have completed their training and have some experience in the field. “Advanced” means that the approach requires a high level of experience and skill and is typically performed by highly trained and experienced surgeons who have mastered the technique. The level of experience required may vary depending on the complexity of the procedure and the specific challenges and risks involved. For a clearer understanding and comparison of these approaches, we provide an overview of the indications, contraindications, benefits, drawbacks and potential advancements of these methods [10,21,22,23,24,25,26,27,28,29,30,31,32,33,34,35,36,37,38].

**Table 2 cancers-15-02931-t002:** The utility of autofluorescence near-infrared imaging in identification of parathyroid glands during thyroidectomy or/and parathyroidectomy.

Paper	Procedure	Dataset	Sensitivity (%)	Specificity (%)	Other (%)	Autofluorescence of PGs	Time for PG Identification by NIFI
Kim S W 2016 [104]	Thyroidectomy	16 PGs 8 patients	W			Mean P/B 2.76 P/B 1.95–5.20	Before operation 5–10 min During operation <1 min
Kim S W 2017 [105]	Thyroidectomy	70 PGs 38 patients	100	100	PPV: 100 NPV: 100 AC: 100	Mean P/B 4.78 P/B 1.96–12.21	Before operation 10 min During operation 2–3 min
Ladurner R 2018 [106]	Thyroidectomy	41 PGs 20 patients	90				Extra 10 min of operating time
Takahashi T 2020 [107]	Thyroidectomy	41 PGs 36 patients	82.9				
Kose E 2019 [111]	Thyroidectomy Parathyroidectomy	971 specimens 310 patients	98			Mean P/B 1.73	During operation <1 min
		550 specimens additionally sent to pathology	98.5	97.2	PPV: 95.1 NPV: 99.1 AC: 97.6		
Falco J 2016 [112]	Thyroidectomy Parathyroidectomy	28 patients				Mean 40.6 14.1–67.1	
McWade WM 2016 [113]	Thyroidectomy Parathyroidectomy	264 PGs 137 patients	97			P/B 1.2–29	Approximately extra 3–4 min of operating time
Kahramangil B 2018 [114]	Thyroidectomy Parathyroidectomy	594 PGs 210 patients	98				Approximately extra 4 min of operating time
Kim D H 2021 * [115]	Thyroidectomy Parathyroidectomy	17 studies with 1198 participants	96.93	92.48	NPV: 95.17 PPV: 94.88		

PG—parathyroid gland, PPV—positive predictive value, NPV—negative predictive value, AC—accuracy, P/B—parathyroid to background fluorescence ratio, *—systematic review and meta-analysis.

**Table 3 cancers-15-02931-t003:** The utility of indocyanine green fluorescence in identification and preservation of parathyroid glands during thyroidectomy and/or parathyroidectomy.

Paper	Procedure	Dataset	Detected PGs with ICGF (%)	Transient HP (%)	Permanent HP (%)	Incidental PTX (%)	Mean PTH Level (pg/dL)
Zaidi N 2016 [116]	Thyroidectomy	85 PGs 27 patients	84				Patients with at least 2 PGs with the ICGF < 30% of the PG’s volume, POD 1: 9
							Patients with fewer than 2 PGs with the ICGF < 30% of the PG’s volume, POD 1: 19.5
							*p* = 0.05
Yu H W 2016 [117]	Thyroidectomy	44 patients without ICGF		18.2	2.3	15.9	
		22 patients with ICGF		18.2	4.5	0	
Rudin A 2019 [119]	Thyroidectomy	124 patients without ICGF					21
		86 patients with ICGF	82				19
Zaidi N 2016 [120]	PTX	112 PGs 33 patients	92.9				

PTX—parathyroidectomy, PG—parathyroid gland, HP—hypoparathyroidism, ICGF—indocyanine green fluorescence, PTH—parathyroid hormone, POD—postoperative day.

**Table 4 cancers-15-02931-t004:** Preoperative risk factors for postoperative bleeding.

Paper	Risk Factors
Scaroni M 2020 [130]	Graves’ disease, benign pathology, previous thyroid surgery, age > 65 years, African American race, history of alcohol abuse, BMI greater than >30 kg/m, male sex
Saitou M 2021 [131]	Graves’ disease, advanced age, male sex, use of anticoagulants, complications of haematological diseases (inherited bleeding disorder characterised by the absence or reduced levels of clotting factor VIII or IX)
Wojtczak B 2018 [129]	Graves–Basedow disease, toxic adenoma, toxic multinodular goitre
Weiss A 2014 [132]	Age > 45 years, African American and Native American patients, male sex, inflammatory conditions of the thyroid, chronic kidney disease
Edafe O 2020 [127]	Thyroidectomy (vs. hemithyroidectomy and parathyroidectomy), older age, male gender, antithrombotic medication, bleeding disorders
Sun N 2020 [128]	BMI > 30 kg/m, age > 45 years, partial thyroidectomy, inflammatory thyroid disease, bleeding disorders, chronic kidney diseases, hypertension, diabetes
Doran HE 2021 [124]	Retrosternal goitre, hyperthyroidism at presentation, male sex, total thyroidectomy, patient age
Patel KN 2020 [123]	Male sex, age > 65 years, smoking, continued use of antiplatelet or anticoagulant medications, Graves–Basedow disease, total thyroidectomy (vs. lobectomy), drain placement, African American race, history of alcohol abuse, 3 or more significant comorbidities, substernal thyroidectomy, reoperation
Liu J 2017 [133]	Older age, male sex, Graves’ disease, antithrombotic agents use, bilateral operation, previous thyroid surgery

**Table 5 cancers-15-02931-t005:** Development of the acronym ITSRED FRED.

I	Irrigation of wound with water—to clear and search for bleeding
T	Time—wait for some time
S R	Systolic blood pressure > 100 mm Hg at closing Relieve neck extension—for example, flexing the neck
E D	Elevate venous pressure—using the Valsalva manoeuvre or by positioning the patient with their head down Do not drain—do not drain it; deal with it and consider sealing agents, e.g., fibrin
FRED	Flaps—evaluation of bleeding under subplatysmal flaps prior to closure

## Data Availability

The datasets used and/or analysed during the current study are available from the corresponding author upon reasonable request.

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
