# Peer review of "Modern Surgical Techniques of Thyroidectomy and Advances in the Prevention and Treatment of Perioperative Complications"

_cancers, 2023, doi:10.3390/cancers15112931_

Round 1

Reviewer 1 Report

The authors have done an excellent job summarizing the advancements in thyroid surgery and identifying key points from the extensive literature. However, there are a few concerns that need to be addressed.

Major

1. In Table 1, it is inappropriate to list subcutaneous emphysema and pneumothorax as the main complications of the trans-axillary approach (TAA). When referring to TAA, it is important to note that the gasless version developed by Korean surgeons is the mainstay, and there are more relevant studies published on this method than the gas insufflation version. Subcutaneous emphysema and pneumothorax are not the main complications of gasless TAA. On the contrary, subcutaneous emphysema is a common complication (usually minor in severity and self-limited) of techniques that require CO2 insufflation, such as BABA and TOETVA.

2. In line 803-806, the authors mention that TOETVA has a lower rate of perioperative bleeding than classical surgical methods. However, evidence to support this claim is needed. Dr. Anuwong's study published in JAMA Surgery found no difference in bleeding risk between TOETVA and open thyroidectomy. Additionally, bleeding is the most common cause of TOETVA conversion to open surgery, as observed in Dr. Anuwong's study and many others.

Anuwong A, Ketwong K, Jitpratoom P, Sasanakietkul T, Duh QY. Safety and Outcomes of the Transoral Endoscopic Thyroidectomy Vestibular Approach. JAMA Surg. 2018 Jan 1;153(1):21-27.

3. In line 915-919, the authors reference Zhao et al. (reference 150) and state that "With endoscopic surgery, CIONM with percutaneous stimulation is not possible." However, it should be noted that CIONM is feasible in TOETVA with a stimulating electrode placed directly on the vagus nerve. Therefore, this statement should be revised.

Chen HK, Chen CL, Wen KS, Lin YF, Lin KY, Uen YH. Application of transoral continuous intraoperative neuromonitoring in natural orifice transluminal endoscopic surgery for thyroid disease: a preliminary study. Surg Endosc. 2018 Jan;32(1):517-525.

4. In line 1104-1109, the authors conclude that new approaches (TAA/BABA/TOETVA) are less invasive, reduce hospital stay, may have lower hospital costs, and decrease nosocomial infections when compare with conventional open surgery. However, these statements are controversial. First, many surgeons do not consider these new approaches as minimally invasive due to the larger skin flap, especially in the case of TAA and BABA. Second, hospital stays are usually similar between endoscopic/robotic surgery and conventional surgery. Finally, the cost is typically higher for endoscopic/robotic surgery due to the cost of endoscopic instruments, and operative times are typically longer than in open surgery. Therefore, this paragraph should be revised or supported by concrete evidence.

Minor:

1. In line 198, PEC II stands for "Pectoralis nerve block II" in the original article (reference 51), not "pectoralis major II."

2. In line 669, the author's name of the cited reference (reference 114) is missing. "Zaidi et al. studied the utility of..."

Author Response

Dear Sir/Madam,

Thank You very much for Your feedback and guidance. They were very valuable for us and allowed us to draw attention to aspects that may have previously remained undeveloped. We have tried to take them to heart and improve our review on the basis of Your recommendations. We hope that as a result of these corrections, our review will be able to convey the subject matter of the issue to the readers of "Cancers" in the most appropriate and complete way. Below there are the comments on the revisions we have made. Thank You again for taking the time to read our manuscript.

Yours faithfully,

Authors

1. In Table 1, it is inappropriate to list subcutaneous emphysema and pneumothorax as the main complications of the trans-axillary approach (TAA). When referring to TAA, it is important to note that the gasless version developed by Korean surgeons is the mainstay, and there are more relevant studies published on this method than the gas insufflation version. Subcutaneous emphysema and pneumothorax are not the main complications of gasless TAA. On the contrary, subcutaneous emphysema is a common complication (usually minor in severity and self-limited) of techniques that require CO2 insufflation, such as BABA and TOETVA.

1. Thank you for your insightful comments regarding our article. We have carefully reviewed your feedback and made the necessary corrections to Table 1. We appreciate your feedback and have made the necessary corrections in Table 1 as per your recommendations.

We acknowledge that it is inappropriate to list subcutaneous emphysema and pneumothorax as the main complications of the TAA, as these complications are usually associated with techniques that require CO2 insufflation, such as BABA and TOETVA. We appreciate your reminder that the gasless version of TAA developed by Korean surgeons is the mainstay, and that there are more relevant studies published on this method than the gas insufflation version. We have incorporated this information into our article to ensure accuracy and clarity.

We thank you again for your comments and hope that the revised article will meet your expectations.

2. In line 803-806, the authors mention that TOETVA has a lower rate of perioperative bleeding than classical surgical methods. However, evidence to support this claim is needed. Dr. Anuwong's study published in JAMA Surgery found no difference in bleeding risk between TOETVA and open thyroidectomy. Additionally, bleeding is the most common cause of TOETVA conversion to open surgery, as observed in Dr. Anuwong's study and many others.

Anuwong A, Ketwong K, Jitpratoom P, Sasanakietkul T, Duh QY. Safety and Outcomes of the Transoral Endoscopic Thyroidectomy Vestibular Approach. JAMA Surg. 2018 Jan 1;153(1):21-27.

2. Thank you for your review, and we would like to confirm that, following your suggestion, the relevant text fragment will be removed from our article.

After a thorough analysis of the available sources, we have come to the conclusion that it is not possible to substantiate our claim regarding the TOTEVA surgical technique and bleeding in thyroidectomy. We are aware that there are studies that touch on this topic, but they are reported on a very small sample size, which does not allow for a unanimous position on this matter.

We want to assure you that we are committed to ensuring the accuracy and credibility of our publications. Therefore, based on your feedback, we have decided to remove the mentioned text fragment.

3. In line 915-919, the authors reference Zhao et al. (reference 150) and state that "With endoscopic surgery, CIONM with percutaneous stimulation is not possible." However, it should be noted that CIONM is feasible in TOETVA with a stimulating electrode placed directly on the vagus nerve. Therefore, this statement should be revised.

Chen HK, Chen CL, Wen KS, Lin YF, Lin KY, Uen YH. Application of transoral continuous intraoperative neuromonitoring in natural orifice transluminal endoscopic surgery for thyroid disease: a preliminary study. Surg Endosc. 2018 Jan;32(1):517-525.

3. Indeed, originally this important aspect was omitted in our work and we are really grateful for noticing it. The issue of the use of CIOMT in endoscopic surgery has been studied by us in more detail. We have added a short excerpt describing the possibility of using CIOMT in TOETVA. Thank You for Your advice.

4. In line 1104-1109, the authors conclude that new approaches (TAA/BABA/TOETVA) are less invasive, reduce hospital stay, may have lower hospital costs, and decrease nosocomial infections when compare with conventional open surgery. However, these statements are controversial. First, many surgeons do not consider these new approaches as minimally invasive due to the larger skin flap, especially in the case of TAA and BABA. Second, hospital stays are usually similar between endoscopic/robotic surgery and conventional surgery. Finally, the cost is typically higher for endoscopic/robotic surgery due to the cost of endoscopic instruments, and operative times are typically longer than in open surgery. Therefore, this paragraph should be revised or supported by concrete evidence.

4. Thank you very much for this accurate comment. Indeed, the conclusions were too generalized. After revising the article, we clarified in the conclusions which benefits apply to specific methods. All conclusions are from the articles we cited in the relevant sections of the paper.

5. In line 198, PEC II stands for "Pectoralis nerve block II" in the original article (reference 51), not "pectoralis major II."

5. We appreciate your attention to detail and would like to address your concern regarding our use of the term "pectoralis major II”.

After reviewing our sources, we have confirmed that PEC II stands for "Pectoralis nerve block II," as stated in reference 51. We apologize for any confusion this may have caused and have made the necessary correction in our article to reflect the accurate term.

6. In line 669, the author's name of the cited reference (reference 114) is missing. "Zaidi et al. studied the utility of..."

6. Thank you for noticing this mistake. We have added the name of the author.

Reviewer 2 Report

hello

thank you for an interesting paper

the aim of study needs to be more clarified

in the introduction chapter main goal of the paper should be presented in 1-2 brief sentences

rest of the goals and important clinical and surgical data should be addressed in a separate line before the main goal

limitations of each surgical approach should be addressed

I'm missing a direct chart flow diagram, why only 123 papers were used in this study, what were the inclusion/exclusion criteria - please make a diagram

paragraph 3.3.2 is missing the author - et al? 

when describing each approach, I would also add their advantages and disadvantages

is the neuronavigation always used in all type of presented surgical approaches?

is there any relation between surgical field draping, surgical approach and placement of the electrodes for the  neuromonitoring?

some sentences are too long and hard to read - please re-arrange them

thank you for the intersting paper

Author Response

Dear Sir/Madam,

Thank You very much for Your feedback and guidance. They were very valuable for us and allowed us to draw attention to aspects that may have previously remained undeveloped. We have tried to take them to heart and improve our review on the basis of Your recommendations. We hope that as a result of these corrections, our review will be able to convey the subject matter of the issue to the readers of "Cancers" in the most appropriate and complete way. Below there are the comments on the revisions we have made. Thank You again for taking the time to read our manuscript.

Yours faithfully,

Authors

1. the aim of study needs to be more clarified

in the introduction chapter main goal of the paper should be presented in 1-2 brief sentences

rest of the goals and important clinical and surgical data should be addressed in a separate line before the main goal

1. Thank you very much for this accurate comment. We have tried to formulate the purpose of the work more clearly.

2. limitations of each surgical approach should be addressed

when describing each approach, I would also add their advantages and disadvantages

2. We truly appreciate your insightful comments, which have helped us to improve the quality of our work.

In response to your suggestions, we have added a section on the limitations of each surgical approach and have described their advantages and disadvantages more thoroughly. Overall, we believe that these changes have greatly strengthened the article we believe that these changes have strengthened the manuscript and will provide readers with a more comprehensive understanding of the topic.

3. I'm missing a direct chart flow diagram, why only 123 papers were used in this study, what were the inclusion/exclusion criteria - please make a diagram

3. Thank you for this idea. We have added a diagram to show the process of including papers in our systematic review. We hope this will make the criteria for inclusion of papers in the review clearer for readers.

4. paragraph 3.3.2 is missing the author - et al? 

4. Thank you for noticing this mistake. We have added the name of the author.

5. is the neuronavigation always used in all type of presented surgical approaches?

5. Thank You very much for asking this important question. We have deeply studied the international standards guideline statement. Based on it, we added a few-sentence excerpt addressing the topic of surgical approaches.

6. is there any relation between surgical field draping, surgical approach and placement of the electrodes for the  neuromonitoring?

6. Thank You for raising this topic. We have done research to look for any information about factors that affect the position of electrodes. Unfortunately, at this point, there is a lack of studies that address these issues. Indeed, the placement of electrodes may be in correlation with variables such as surgical draping and differences in surgical approaches. However, since we do not have access to this type of information, we cannot include it in our paper. 

7. some sentences are too long and hard to read - please re-arrange them

7. Thank you for your review of our article, and we are glad to hear that you noticed potential difficulties in understanding some parts of the text. We have worked to simplify and improve our article to make it more understandable and readable for our readers.

We appreciate your feedback and suggestions, as they have allowed us to make concrete changes to the text. We have revised long sentences to make them more comprehensible.

We hope that the revisions we made will lead to a better understanding of our article. Thank you for your time and willingness to help improve our scholarly work.

Round 2

Reviewer 2 Report

thank you very much

the authors addressed all issues according to my hints

the paper can be send for further proceedings